# MicroRNA, mRNA, and Proteomics Biomarkers and Therapeutic Targets for Improving Lung Cancer Treatment Outcomes

**DOI:** 10.3390/cancers15082294

**Published:** 2023-04-14

**Authors:** Qing Ye, Rebecca Raese, Dajie Luo, Shu Cao, Ying-Wooi Wan, Yong Qian, Nancy Lan Guo

**Affiliations:** 1West Virginia University Cancer Institute, West Virginia University, Morgantown, WV 26506, USA; qiye@mix.wvu.edu (Q.Y.); rebecca.raese@gmail.com (R.R.); luodajie@hotmail.com (D.L.); caoshu1988@gmail.com (S.C.); wywooi@yahoo.com (Y.-W.W.); 2Health Effects Laboratory Division, National Institute for Occupational Safety and Health, Morgantown, WV 26505, USA; yaq2@cdc.gov; 3Department of Occupational and Environmental Health Sciences, School of Public Health, West Virginia University, Morgantown, WV 26506, USA

**Keywords:** lung cancer, diagnosis, prognosis, CRISPR-Cas9/RNAi, radiotherapy response, drug response, repositioning drugs

## Abstract

**Simple Summary:**

This study identified a set of 73 microRNAs (miRNAs) that can accurately detect lung cancer tumors from normal lung tissues. Based on the consistent expression patterns associated with patient survival outcomes and in tumors vs. normal lung tissues, 10 miRNAs were considered to be putatively tumor suppressive and 4 miRNAs were deemed as oncogenic in lung cancer. From the list of genes that were targeted by the 73 diagnostic miRNAs, DGKE and WDR47 had significant associations with responses to both systemic therapies and radiotherapy in lung cancer. Based on our identified miRNA-regulated network, we discovered three drugs—BX-912, daunorubicin, and midostaurin—that can be repositioned to treat lung cancer, which was not known before.

**Abstract:**

The majority of lung cancer patients are diagnosed with metastatic disease. This study identified a set of 73 microRNAs (miRNAs) that classified lung cancer tumors from normal lung tissues with an overall accuracy of 96.3% in the training patient cohort (*n* = 109) and 91.7% in unsupervised classification and 92.3% in supervised classification in the validation set (*n* = 375). Based on association with patient survival (*n* = 1016), 10 miRNAs were identified as potential tumor suppressors (hsa-miR-144, hsa-miR-195, hsa-miR-223, hsa-miR-30a, hsa-miR-30b, hsa-miR-30d, hsa-miR-335, hsa-miR-363, hsa-miR-451, and hsa-miR-99a), and 4 were identified as potential oncogenes (hsa-miR-21, hsa-miR-31, hsa-miR-411, and hsa-miR-494) in lung cancer. Experimentally confirmed target genes were identified for the 73 diagnostic miRNAs, from which proliferation genes were selected from CRISPR-Cas9/RNA interference (RNAi) screening assays. Pansensitive and panresistant genes to 21 NCCN-recommended drugs with concordant mRNA and protein expression were identified. DGKE and WDR47 were found with significant associations with responses to both systemic therapies and radiotherapy in lung cancer. Based on our identified miRNA-regulated molecular machinery, an inhibitor of PDK1/Akt BX-912, an anthracycline antibiotic daunorubicin, and a multi-targeted protein kinase inhibitor midostaurin were discovered as potential repositioning drugs for treating lung cancer. These findings have implications for improving lung cancer diagnosis, optimizing treatment selection, and discovering new drug options for better patient outcomes.

## 1. Introduction

Lung cancer is currently the leading cause of cancer-related deaths in both men and women, accounting for approximately 27% of all deaths attributed to cancer. Additionally, lung cancer is the second most commonly occurring cancer in both sexes. Non-small cell lung cancer (NSCLC) comprises 80–85% of all lung cancer cases. Survival statistics for patients with lung cancer vary depending on the stage of cancer at diagnosis, with local and distant recurrence comprising the major causes of treatment failure. Another challenge in the treatment lies in the fact that most lung cancers are only diagnosed when the disease has already reached an advanced stage, as early stage lung cancer is often asymptomatic. Early diagnosis of lung cancer would serve to improve patient survival rates and response to therapies. Research has shown that the 5-year overall survival was markedly greater for lung cancers diagnosed in the first 2 years of screening (77%) when compared to cancers that were discovered in the third to the fifth year of screening (36%) [1].

MicroRNA (miRNA) expression profiles may provide useful information concerning tumor development and thus could serve as important diagnostic biomarkers for early disease detection. Since their discovery in 1993, miRNAs have initiated a vast amount of research not only concerning their nature but also their roles in the regulation of a multitude of biological processes to which they are attributed. The specific functions and mechanisms of miRNA have been extensively reviewed [2,3]. Briefly, miRNAs are single-stranded, non-coding RNA molecules that average 22 nucleotides in length. They are responsible for controlling gene expression by acting as post-transcriptional regulators. In their mature form, miRNAs combine with proteins from the Argonaut family to form the RNA-induced silencing complex (RISC). This complex then binds to the 3′ untranslated region of various target genes and prevents their expression through either translational repression or mRNA degradation. Because each miRNA can act upon several target genes, their overall effect on the abundance of various proteins is magnified. Furthermore, miRNAs can modulate multiple points in any disease pathway. Therefore, even minor alterations in miRNA expression can result in substantial effects on protein abundance and, thus, biological processes and signaling pathways [3,4]. MiRNA expression patterns and their implications in carcinogenesis have generated extensive interest in the field of cancer biology as potential diagnostic biomarkers and therapeutic targets [5].

MiRNAs have garnered interest as biomarkers in cancer diagnostics and could ultimately prove to be superior to mRNAs in this application [6]. Since miRNAs are more stable than mRNA, they can be successfully isolated from formalin-fixed paraffin-embedded (FFPE) samples as well as minimally non-invasive biological samples, providing an additional asset to their diagnostic utility. The fact that miRNA expression profiles are unique to each tissue and tumor type further supports the importance of their role in cancer diagnostics [4]. The presence of miRNAs in a variety of bodily fluids, such as serum, plasma, saliva, and amniotic fluid [7], is a particularly fascinating but little-understood aspect of miRNA biology. Diagnostic miRNA markers were identified in tissues [1,8], serum [9], blood [10], and sputum samples [11] from NSCLC patients. These studies show promising results for the potential application of miRNA expression profiles as useful early diagnostic tools in a variety of non-invasive biological samples.

Intracellular and exosomal miRNAs play different roles in maintaining cellular homeostasis and intercellular communication. Intracellular miRNAs attach to particular mRNA molecules and prevent their translation within cells [12]. Intracellular miRNAs play a role in a variety of biological functions, including development, differentiation, metabolism, and diseases, including cancer, cardiovascular, and neurodegenerative disorders [13]. In contrast, exosomal miRNAs are enclosed within exosomes, which are tiny vesicles that are released by cells. MiRNAs and other proteins can be transported between cells by exosomes, which are essential for intercellular communication [14]. Exosomal miRNAs control gene expression in target cells and participate in many biological processes, including immune control, angiogenesis, and cancer metastasis [15], and can also serve as diagnostic and prognostic biomarkers for various diseases, including cancer [16], cardiovascular disease, and infectious diseases [17].

The highly conserved family of tissue-specific miRNAs keeps cells in a stable state by negatively regulating gene expression in general. Since intracellular and extracellular miRNAs have a broad range of target genes and affect almost every signaling pathway, from cell cycle checkpoints to cell proliferation to apoptosis, proper regulation of miRNA expression is necessary to maintain normal physiology. Some miRNAs function as tumor suppressors and oncogenes, and their expression is dysregulated in different cancers. Although cancer treatments are currently available to slow the growth and spread of tumors, there are not many effective diagnostic and treatment methods for different cancers. Specific miRNA profiling can distinguish molecularly diverse tumors based on their phenotypic characteristics, which can then be used to overcome diagnostic and therapeutic obstacles [18]. The available artificial intelligence/machine learning (AI/ML) tools, bioinformatics resources, and data consortia accelerate the discoveries of miRNA-based theranostics. 

In this study, we sought to identify relevant miRNAs that are differentially expressed in NSCLC patients compared to normal controls and construct an accurate classifier for discriminating lung cancer tumors from normal lung tissues using the patient samples that we collected (*n* = 109). By employing bioinformatics tools and statistical analyses and leveraging public data, we aim to identify a select group of miRNA biomarkers with potential clinical utility in the diagnosis of lung cancer (*n* = 462). From the diagnostic miRNAs, we further identified the prognostic miRNAs with The Cancer Genome Atlas lung adenocarcinoma (TCGA-LUAD) and lung squamous cell carcinoma (TCGA-LUSC) patient data (*n* = 1016). We then identified the target genes of these miRNAs with TarBase. We further investigated the performance of these targeted mRNAs/proteins in terms of prognosis, proliferation, and drug sensitivity. Proliferative genes were assessed using public CRISPR-Cas9 screening data in 94 human non-small cell lung cancer (NSCLC) cell lines and RNA interference (RNAi) screening data in 92 human NSCLC cell lines in the Cancer Cell Line Encyclopedia (CCLE). The drug response was analyzed using the public data in 135 human NSCLC cell lines in CCLE. Next, new compounds for treating NSCLC were determined with Connectivity Map (CMap) based on the gene expression signatures in tumorigenesis, patient survival, and proliferation. CMap provides a valuable resource for drug discovery and repositioning efforts. It allows the identification of compounds that induce similar or opposite transcriptional profiles to a query signature, thereby providing insights into potential drug mechanisms of action and novel indications for existing drugs. Finally, the NSCLC responders of the new compounds were identified by analyzing the CCLE profiles of drug responses. The overall flowchart was provided in Appendix A.

## 2. Materials and Methods

### 2.1. Patient Samples

A total of 117 patient tissue samples were procured from the WVU Tissue Bank (Morgantown, WV, USA) and the Cooperative Human Tissue Network (CHTN) and stored at –80 °C. These samples included 109 snap-frozen lung tumor samples, as well as 22 matched, normal lung tissue samples. The samples were examined by pathologists before being cataloged in the biorepositories at WVU tissue bank and CHTN. For tumor tissue samples, it was certified that the tumor content is at least 50% in each sample. For non-cancerous normal lung tissues, it was confirmed that no tumor was present in the sample. This information was provided in the pathology reports of the tissue samples that we received. Before our experiments, a pathologist’s assistant further examined the tissue samples. No experimental control was carried out to confirm that there were no tumor cells in the normal tissues. Blood samples from 4 lung cancer patients and 6 normal individuals were provided by CHTN and were sent frozen on dry ice following collection in EDTA blood tubes. These samples were also stored at –80 °C until use. Pathology reports were provided with all samples to verify cancer stage, tumor grade, and histology. All identifying notations were removed from the pathology reports prior to delivery to ensure patient privacy was maintained. This study was approved with an Institutional Review Board exemption from West Virginia University. This cohort (MBRCC/CHTN) was used as the training set to identify diagnostic miRNA markers (Table 1).

A validation cohort (*n* = 375) was retrieved from the NCBI Gene Expression Omnibus database with accession number (GSE15008). This cohort contains 187 lung cancer tumor tissue samples and 188 normal lung tissue samples, most of which were from adjacent normal match tissues.

The Cancer Genome Atlas (TCGA) is a publicly accessible database that provides a comprehensive resource for genomic data on various types of cancer. For this study, we obtained miRNA expression, gene expression, and mutation data for TCGA lung adenocarcinoma (TCGA-LUAD; http://linkedomics.org/data_download/TCGA-LUAD/, accessed on 10 January 2023) and TCGA lung squamous cell carcinoma (TCGA-LUSC; http://linkedomics.org/data_download/TCGA-LUSC/, accessed on 10 January 2023) from LinkedOmics [19]. We included a total of 63 LUAD and 136 LUSC normalized miR-gene level samples from the Illumina GenomeAnalyzer platform, as well as 450 LUAD and 342 LUSC normalized miR-gene level samples from the Illumina HiSeq platform. The RNA sequencing data for LUAD (*n* = 515) and LUSC (*n* = 501) was measured using the Illumina HiSeq 2000 RNA Sequencing platform. The miRNA and gene expression data were log_2_ transformed. Patient clinical information was obtained from LinkedOmics.

We also conducted a prognostic evaluation in non-small cell lung cancer (NSCLC) using RNA sequencing data from Xu et al. [20]. Log_2_-transformed mRNA data of 51 LUAD tumors and 49 paired NATs used in this study was obtained from Xu’s LUAD cohort.

### 2.2. RNA Isolation and Quality Assessment

Total RNA was extracted from frozen tumor and normal tissue samples using the mirVana miRNA Isolation kit and following the manufacturer’s protocol (Ambion, Austin, TX, USA). Total RNA was isolated from frozen blood samples using a modified PAXgene protocol that was detailed by Beekman et al. [21]. Briefly, frozen blood samples were thawed on ice and transferred to PAXgene blood collection tubes. Extraction of total RNA was performed using the PAXgene Blood miRNA kit according to the manufacturer’s instructions (Qiagen, Valencia, CA, USA). RNA concentration was determined using the NanoDrop 1000 Spectrophotometer (NanoDrop Technologies, Wilmington, DE, USA), and RNA quality was assessed using the 2100 Bioanalyzer (Agilent Technologies, Santa Clara, CA, USA). RNA quality was checked with UV Spectrometry (260/280 > 1.8) and RNA quality determination gel (1% agarose–2% formaldehyde QC gel). In all, total RNA extracted from 109 patient tissue samples and 10 patient blood samples met the quality control criteria and were selected for further analysis.

### 2.3. Microarray Analysis

MiRNA profiling, including additional quality controls, was completed by Ocean Ridge Biosciences (Palm Beach Gardens, FL, USA) using custom microarrays containing 1087 human miRNA probes. These miRNA arrays incorporate all 1098 human miRNAs present in the Sanger Institute mirBASE version 15. Quality control features that were included in this analysis consist of negative controls, specificity controls, and spiking probes. Furthermore, a detection threshold was calculated for each array by summing five times the standard deviation of the background signal and the 10% trim mean of the negative control probes. Using these values, we were able to filter out probes with uniformly low signals prior to completing our statistical analysis. The miRNA microarray data were quantile normalized between arrays. The raw microarray data and patient clinical information are available in NCBI Gene Expression Omnibus with accession number GSE31275.

### 2.4. Hierarchical Clustering Analysis and Heatmap

Hierarchical two-dimensional clustering analyses were performed using the expression profiles of the identified miRNA markers with the Heatplus function in the R package. Similarity metrics were Manhattan distance, and the cluster method was Ward’s linkage. Heatmaps were then generated in the R package 4.2.1.

### 2.5. Nearest Centroid Classification

In the external validation of the lung cancer diagnostic model, the nearest centroid method was used to classify lung cancer samples from normal lung tissues. Specifically, Pearson’s correlation coefficients between a new patient’s miRNA expression profiles and those of the lung cancer centroid and the normal centroid in the training cohort were computed, respectively. A patient sample was predicted as lung cancer if the correlation with the lung cancer centroid was greater than that with the normal centroid; otherwise, it was predicted as normal.

### 2.6. MiRNA Targeted Genes

TarBase [22] is a comprehensive database that provides a centralized resource for information on experimentally validated microRNA-target interactions. The bulk download modules of TarBase v7.0 (http://diana.imis.athena-innovation.gr/DianaTools/data/TarBase7data.tar.gz, accessed on 10 January 2023) and v8.0 (https://dianalab.e-ce.uth.gr/html/diana/web/index.php?r=tarbasev8%2Fdownloaddataform, accessed on 10 January 2023) were obtained, experimentally confirmed target genes of selected miRNAs were retrieved.

### 2.7. Cancer Cell Line Encyclopedia

We obtained RNA sequencing data of 135 cell lines, 108,344 mutations, and 4223 fusions for NSCLC from the Cancer Cell Line Encyclopedia (CCLE) release DepMap Public 22Q2 [23] (https://depmap.org/portal/download/all/, accessed on 25 January 2023). In addition, from the CCLE 2019 release [24] (https://depmap.org/portal/download/all/, accessed on 25 January 2023), miRNA data of 123 NSCLC cell lines were collected. From the study conducted by the Gygi lab [25] (https://gygi.hms.harvard.edu/publications/ccle.html, accessed on 26 January 2023), we acquired proteomic data for 63 NSCLC cell lines. The mRNA and proteomic data are both log_2_-tranformed. Furthermore, the mean of protein expression was centered at 0. These datasets provide a comprehensive resource for molecular profiles of NSCLC cell lines, enabling the investigation of the molecular characteristics and potential therapeutic targets of NSCLC.

### 2.8. Drug Response

The PRISM (Profiling Relative Inhibition Simultaneously in Mixtures) framework [26] is a computational tool that estimates drug sensitivities of cancer cell lines using molecular profiling data, particularly gene expression data from the CCLE database. We obtained the PRISM repurposing dataset of the secondary screen from DepMap release PRISM Repurposing 19Q4 (https://depmap.org/portal/download/all/, accessed on 25 January 2023). For this study, we used the PRISM drug response data from 94 NSCLC cell lines for 1448 compounds to investigate the drug sensitivity of specific genes.

The Genomics of Drug Sensitivity in Cancer (GDSC) [27] database is a publicly available resource that provides comprehensive data on drug responses in a large panel of cancer cell lines. The drug screening data for this study were obtained from the project website (https://www.cancerrxgene.org/, accessed on 25 January 2023). We used drug response data from 78 NSCLC CCLE datasets from GDSC1 and 69 NSCLC CCLE datasets from GDSC2. Details of the method used to categorize each cell line’s drug sensitivity were previously published by our lab [28,29].

### 2.9. Proliferation Assays

Gene knockout effects in CCLE using CRISPR-Cas9 screens [30] were quantified in Project Achilles and available from DepMap Public 22Q2 release (https://depmap.org/portal/download/all/, accessed on 25 January 2023). The gene-level CRISPR-Cas9 dependency scores were standardized using the CERES method. For genome-scale screening data of RNA interference (RNAi) knockdown [31] for NSCLC cell lines, the Project Achilles dataset was accessed via the website of the DepMap portal (https://depmap.org/R2-D2/, accessed on 25 January 2023). For each cell line, the average gene dependency scores were estimated with the DEMETER2 v6 algorithm.

To determine the impact of gene knockout in NSCLC cell lines, we analyzed genome-wide CRISPR-Cas9 knockout dependency scores for 94 NSCLC cell lines and RNAi screening data for 92 NSCLC cell lines. Genes were classified as essential or non-essential based on their importance to cell growth in each line. For the normalization of the gene effects in each cell line, the median knockout effect value of the essential gene was set to –1, and that of the non-essential gene was set to 0. A normalized dependence score of less than –0.5 was considered to have a significant effect on CRISPR-Cas9 knockout and RNAi knockdown.

### 2.10. Connectivity Map (CMap)

Connectivity Map [32,33] (CMap; https://clue.io/, accessed on 25 January 2023) is a public database that contains gene expression profiles of human cells treated with various bioactive small molecules. In this study, we identified gene expression signatures and discovered their potential connected repositioning drugs by utilizing the CMap web tool. We considered connectivity scores to be significant if they exceeded 0.9 and had a *p*-value of 0.05 or below.

### 2.11. Statistical Analysis

The Significance Analysis of Microarrays (SAM) method [34] was used to identify miRNA markers exhibiting differential expression in lung cancer vs. normal samples. The following cut-off values were selected to evaluate those miRNA markers that demonstrated an expression fold change of >2 or <0.5 between normal and tumor tissues or between blood samples from normal or cancer patients; *p <* 0.05; false discovery rate (FDR) < 0.05, using unpaired *t*-tests, or paired *t*-tests based on the sample set. Student’s *t*-tests were used to compare the gene expression of the two groups, and a two-sided *p*-value of 0.05 or below was deemed statistically significant. We conducted survival analysis using the Kaplan–Meier method with the survival package in R. To evaluate differences in survival probabilities across groups, we performed log-rank tests on the Kaplan–Meier curves. All the statistical analyses were performed using Rstudio on R version 4.2.1.

## 3. Results

### 3.1. Identification of Diagnostic miRNA Markers

Using the MBRCC/CHTN cohort (Table 1) as the training set, 66 miRNAs had significant differential expression (*p <* 0.05, unpaired *t*-tests; FDR < 0.05, SAM) with at least a two-fold change (either over-expression or under-expression) in 87 lung cancer samples and 22 normal lung tissues. When 22 lung cancer tumor samples and matched normal lung tissues were analyzed, 58 miRNA showed significant differential expression (*p <* 0.05, unpaired *t*-tests; FDR < 0.05, SAM) with at least a two-fold change. Among these identified miRNA markers, 51 were common in both sets, leading to a set of 73 unique miRNA diagnostic markers (Appendix A).

Using the expression profiles of these 73 miRNA markers, patient samples in the MBRCC/CHTN were separated into two groups with distinct miRNA expression patterns (Figure 1). Specifically, the left panel in Figure 1 contains normal samples with two NSCLC and one carcinoid sample misclassified into this cluster; the right panel in Figure 1 contains lung cancer samples, with one normal tissue misclassified into this cluster. Overall, the overall accuracy of the unsupervised hierarchical clustering was 96.3% (105/109), with a sensitivity of 96.6 (84/87) and a specificity of 95.5% (21/22).

To validate the miRNA-based diagnostic model, both unsupervised and supervised methods were used to classify lung cancer samples from normal lung tissue samples in an external cohort (*n* = 375; GSE15008). Out of 73 identified miRNA markers, 47 were found in the validation set. By using the expression profiles of these 48 miRNA markers, lung cancer samples were separated from the normal lung tissues in unsupervised hierarchical clustering with the same statistical parameters (Figure 2). There were 174/187 (93%) correctly classified cancer samples and 170/188 (90.4%) correctly classified normal tissue samples, giving an overall accuracy of 91.7% in unsupervised classification, indicating that the identified miRNA markers could separate lung cancer and normal lung tissues with distinct expression patterns.

To further validate the potential clinical utility of the miRNA-based diagnostic model, the nearest centroid method was used to predict lung cancer cases in the external validation set. The “lung cancer centroid” and “normal centroid” with the four misclassified samples in the training cohort were used in the classification. Each sample in the GSE15008 cohort was classified based on its miRNA expression correlations with the “lung cancer centroid” and the “normal centroid” in the training set. A patient was predicted as lung cancer if the correlation with the “lung cancer centroid” was greater than that with the “normal centroid”; otherwise, a patient sample was predicted as normal. In general, when the correlation coefficient with the “normal centroid” is less than 0.8, the sample was most likely a lung cancer tumor (Figure 3A). For most misclassified samples, the correlation coefficients with both centroids were very close and hard to distinguish (Figure 3B). The overall accuracy of the lung cancer prediction is 92.3% (346/375), with a sensitivity of 87.2% (163/187) and a specificity of 97.3% (183/188). The miRNA profiles of the training and testing cohorts were based on different miRNA platforms. The highly accurate validation results demonstrate that the miRNA expression-based model could accurately predict lung cancer risk and could be used for diagnostic tests.

### 3.2. Comparison of miRNA Markers in Different Histology

After substantiating the diagnostic capacity of the identified 73 miRNA markers for lung cancer, we sought to investigate their expression patterns in different histology of lung cancer, including NSCLC, small cell lung cancer (SCLC), and carcinoid in the MBRCC/CHTN cohort. A total of 22 miRNA had consistent over-expression in these three histological types (Figure 4A). Among these 22 miRNAs, hsa-miR-210 was significantly over-expressed in SCLC, NSCLC, and carcinoid (*p <* 0.05; unpaired *t*-tests). The other 21 miRNAs were significantly over-expressed (*p <* 0.05; unpaired *t*-tests) in SCLC and NSCLC, but their over-expression was not significant in carcinoid. Notably, hsa-miR-9* had a more than 100-fold over-expression in SCLC (*p <* 0.05; unpaired *t*-tests). A total of 35 miRNAs were under-expressed in SCLC, NSCLC, and carcinoid (Figure 4B). All of these 35 miRNAs were significantly under-expressed in SCLC and NSCLC, but their under-expression was not significant in carcinoid (*p <* 0.05; unpaired *t*-tests). A total of 16 miRNAs had different expression patterns in SCLC, NSCLC, and carcinoid (Figure 4C). Specifically, hsa-miR-3065-3p and hsa-miR-338-5p was over-expressed in carcinoid but were under-expressed in SCLC and NSCLC (*p <* 0.05; unpaired *t*-tests), indicating that these two miRNA markers could potentially be used for classification of lung cancer histological subtypes.

### 3.3. Confirmation of miRNA Expression Patterns in Multiple Cohorts

The expression patterns of the identified miRNA markers were confirmed with the external validation set. A total of 18 miRNAs had consistent over-expression (*p* < 0.05; unpaired *t*-tests) in lung cancer tumor tissues (Figure 5A), including hsa-miR-130b, hsa-miR-141, hsa-miR-182, hsa-miR-183, hsa-miR-193b, hsa-miR-196a, hsa-miR-205, hsa-miR-21, hsa-miR-210, hsa-miR-301b, hsa-miR-31, hsa-miR-411, hsa-miR-494, hsa-miR-7, hsa-miR-708, hsa-miR-9, hsa-miR-933, and hsa-miR-96.

A total of 25 miRNAs were consistently under-expressed (*p* < 0.05; unpaired *t*-tests) in lung cancer tissue samples in both cohorts (Figure 5B), including hsa-miR-1, hsa-miR-101, hsa-miR-126, hsa-miR-138, hsa-miR-143, hsa-miR-144, hsa-miR-145, hsa-miR-195, hsa-miR-218, hsa-miR-223, hsa-miR-30a, hsa-miR-30b, hsa-miR-30c, hsa-miR-30d, hsa-miR-335, hsa-miR-338-3p, hsa-miR-338-5p, hsa-miR-34b, hsa-miR-34c-3p, hsa-miR-34c-5p, hsa-miR-363, hsa-miR-451, hsa-miR-486-5p, hsa-miR-497, and hsa-miR-99a.

### 3.4. MicroRNA Markers in Blood Samples

We sought to identify miRNA markers in blood samples from lung cancer patients (*n* = 4) and normal individuals (*n* = 6). Seven miRNAs showed significant differential expression (*p <* 0.05; unpaired *t*-tests) in lung cancer versus normal samples (Figure 5C). Specifically, hsa-miR-26a and hsa-miR543 were over-expressed in lung cancer; whereas hsa-miR-200b, hsa-miR-449a, hsa-miR-20a*, and hsa-miR-1973 were under-expressed in lung cancer (Figure 5D). When compared with the expression patterns in the tissue samples in the two patient cohorts, hsa-miR-543 had concordant over-expression in both lung cancer tumor tissues and blood (*p <* 0.05; unpaired *t*-tests). hsa-miR-26a was over-expressed in the lung cancer blood but under-expressed in the tumor tissues, whereas hsa-miR-200b was under-expressed in the blood but over-expressed in the tumor tissues in both clinical cohorts (*p <* 0.05; unpaired *t*-tests).

### 3.5. Identification of Prognostic miRNAs

To demonstrate the prognostic performance of the selected miRNAs (Appendix A), the survival analysis of each miRNA was performed on the TCGA-LUAD and TCGA-LUSC datasets. For each miRNA, we defined a log_2_-transformed expression cutoff to divide the patients into groups of over-expression and under-expression and selected the cutoff point that yielded the smallest *p*-value in the statistical survival analysis between the groups. We categorized miRNAs with a hazard ratio > 1 and a *p*-value < 0.05 and over-expressed in lung cancer tumors (Figure 5A) as potential oncogenic miRNAs, and those with a hazard ratio < 1 and a *p*-value < 0.05 and under-expressed in lung cancer tumors (Figure 3B) as potential tumor suppressive miRNAs (Table 2). There were 14 miRNAs with concordant categorization results in lung cancer tumors vs. TCGA-LUAD, TCGA-LUSC, and TCGA-NSCLC (combined TCGA-LUAD and TCGA-LUSC). Detailed results were provided in Appendix A.

### 3.6. Association between miRNA-Targeted Genes and Responses to Systemic Therapies and Radiotherapy

To demonstrate the clinical relevance and support further investigation of the selected diagnostic miRNAs (Appendix A), we retrieved experimentally confirmed target genes using TarBase. A total of 3139 genes were identified as target genes of our discovered diagnostic miRNAs.

We utilized the CCLE drug screening data, which included 21 drugs recommended by the National Comprehensive Cancer Network (NCCN) for systemic or targeted therapy in the treatment of NSCLC. We aimed to identify genes that were pansensitive or panresistant to the 21 NCCN-recommended drugs in NSCLC among all the selected miRNA and their targeted genes. We defined pansensitive genes/miRNAs as those that exhibited sensitivity or lack of resistance to all 21 studied drugs, and panresistant genes/miRNAs as those that exhibited resistance or lack of sensitivity to all 21 studied drugs. We analyzed RNA sequencing, proteomics, and miRNA profiles of human NSCLC cell lines in the CCLE. Specifically, genes/miRNAs with significantly higher expression (*p* < 0.05; two-sample *t*-tests) in NSCLC cell lines sensitive to a specific drug were classified as sensitive genes/miRNAs, whereas genes/miRNAs with significantly higher expression (*p* < 0.05; two-sample *t*-tests) in NSCLC cell lines resistant to the same drug were classified as resistant genes/miRNAs. In this analysis, we only selected the genes that were consistently pansensitive or panresistant at both mRNA and protein levels and the miRNAs that were pansensitive or panresistant (Table 3). Details of the fusion and mutation information of the pansensitive and panresistant genes in NSCLC cell lines were provided in Appendix A.

We then investigated the association between diagnostic miRNAs-targeted genes (Appendix A) and the radiotherapy response. We analyzed the gene expression levels in patients with stage III or IV who underwent radiotherapy in the TCGA-LUAD and TCGA-LUSC cohorts. Specifically, we compared the mRNA expression levels of these genes between patients with a long survival (>58 months; *n* = 5) and short survival (<20 months; *n* = 20) following radiotherapy. Genes that showed significant differential expression (*p* < 0.05, two-sample *t*-tests) between the two groups were classified as radiotherapy-sensitive (higher expression in the long-survival group) or radiotherapy-resistant (higher expression in the short-survival group). We identified 210 genes that were either radiotherapy-sensitive or -resistant. Detailed results were provided in Appendix A.

DGKE and WDR47 were found with significant associations with responses to both systemic therapies and radiotherapy (Table 3, Figure 6 and Figure 7). DGKE is targeted by hsa-miR-3065-5p, and its expression was associated with sensitivity to erlotinib and radiotherapy. Hsa-miR-139-5p targets WDR47, which was resistant to afatinib, brigatinib, and osimertinib and was sensitive to radiotherapy.

### 3.7. Identification of Prognostic miRNA-Targeted Genes

We used Xu’s LUAD dataset, as well as TCGA-LUAD and TCGA-LUSC data, to assess the prognostic significance of the identified miRNA-targeted genes. Genes that were significantly over-expressed in tumor samples in the log_2_-transformed mRNA data or had a significant hazard ratio > 1 in the univariate Cox model were identified as hazardous genes. Conversely, genes that were significantly under-expressed in lung cancer tumor tissues in the log_2_-transformed mRNA data or had a significant hazard ratio < 1 in the univariate Cox model were identified as protective genes. To be selected as mRNA prognostic genes, the genes had to be concordant in at least two of the five categories (mRNA differential expression between tumors vs. non-cancerous adjacent tissues in Xu’s LUAD, as well as significant association with patient survival in Xu’s LUAD, TCGA-LUAD, TCGA-LUSC, and TCGA-NSCLC). Detailed results are provided in Appendix A.

### 3.8. Discovery of Repositioning Drugs

To discover connected perturbagen signatures and repositioning drugs for treating NSCLC, we utilized the above-identified prognostic genes and proliferation genes in CRISPR-Cas9/RNAi screening assays from the miRNA-targeted genes using CMap. Two sets of genes were defined as CMap input lists. CMap input list 1 included (1) an up-regulated gene list (*n* = 100) consisting of protective miRNA-targeted genes (defined in Section 3.7), which also had a significant effect in less than 50% of the tested NSCLC cell lines in both CRISPR-Cas9 and RNAi screening assays, and (2) a down-regulated gene list (*n* = 35) that comprised of hazardous miRNA-targeted genes (defined in Section 3.7) that had a significant effect in more than 50% of the tested NSCLC cell lines in both proliferation screening assays. CMap input list 2 included an up-regulated gene list (*n* = 85) consisting of protective miRNA-targeted genes that had no significant effect in both proliferation screening assays in NSCLC cell lines and a down-regulated gene list (*n* = 79) consisting of hazardous miRNA-targeted genes that had a significant effect in more than 50% of the NSCLC cell lines in at least one of CRISPR-Cas9 and RNAi screening assays. Detailed information on CMap inputs and outputs was provided in Appendix A.

Three targeted therapeutic candidates were selected with a low averaged IC_50_ and EC_50_ in human NSCLC cell lines (Figure 8), indicating their efficacy in inhibiting NSCLC cell growth. The selected compounds were BX-912 from drug set PYRUVATE_DEHYDROGENASE_KINASE_INHIBITOR, daunorubicin from drug set RNA_SYNTHESIS_INHIBITOR, and midostaurin from drug set CP_FLT3_INHIBITOR. These three pathways were not identified in our previous network studies for repositioning drug discovery [28,35,36] (Appendix A).

BX-912 is an inhibitor of 3-phosphoinositide-dependent protein kinase 1 PDK1/Akt signaling [37]. It is an experimental drug that has been studied for its potential therapeutic use in the treatment of breast, colon, and prostate cancers, as well as gliomas [38] and mantle cell lymphoma [39]. Daunorubicin, belonging to the anthracyclines class, is a chemotherapy drug for treating leukemias and Kaposi’s sarcoma [40,41]. Midostaurin is a protein kinase inhibitor. It has been approved for use in the treatment of certain types of blood cancers, including acute myeloid leukemia (AML) and myelodysplastic syndrome (MDS) [42,43]. Their potential utility in treating NSCLC was not known before.

To characterize responders of these three potential repositioning drugs, genes associated with the drug response in CCLE NSCLC cell lines were selected on a genome-wide scale (Table 4). The selected genes had a concordant significantly differential mRNA and protein expression in sensitive vs. resistant NSCLC cell lines to the specific compound in the PRISM data. None of the genes found had a significant association with radiotherapy response. Most of the genes had gene mutations and/or fusions in NSCLC cell lines. Detailed mutation and fusion information associated with the response to the three potential repositioning drugs was provided in Appendix A.

## 4. Discussion

According to the National Institute of Cancer (NCI) data, about 55% of lung cancer cases are diagnosed when the disease has already metastasized [44]. The National Lung Screening Trial (NLST) conducted a study of 53,454 participants at high risk for lung cancer in the United States. The NLST results showed that three annual computed tomographic (CT) screenings led to a 20% lower mortality from lung cancer than screening with chest radiography after a median follow-up of 6.5 years [45]. Nevertheless, CT screening for lung cancer has several limitations, including a high false positive rate and associated overdiagnosis and overtreatment, radiation, limited coverage for specific age and smoking history groups, and costs. According to NLST, The false positive rate of CT screening was about 96.4% [46]. Thus, 96 out of 100 individuals who undergo CT screenings will have a false-positive result showing the presence of a lung nodule or another abnormality that is not cancer. As a consequence, it was reported that 9% of the patients who received lung cancer surgery did not have malignancy [47]; the benign rates in invasive biopsies for patients of suspicious lung cancer range from 20–30% [48,49]. Hence, the development of minimally invasive blood-based diagnostic tests is an unmet clinical need to improve clinical decision-making in lung cancer treatment.

Lung cancer is a heterogeneous disease with distinct histological subtypes and complex somatic mutations. Tumor heterogeneity within disease sites contributes to resistance to cancer therapies and poses a challenge to biomarker discovery [50,51]. MiRNAs have emerged as important diagnostic and prognostic biomarkers and therapeutic targets in cancer treatment due to their functional involvement in gene and protein regulation and intracellular signaling [52,53]. MiRNA biomarkers are promising for the early detection of NSCLC supplementing low-dose CT screening [54] and the prognosis of advanced NSCLC patients treated with immunotherapy [55]. 

In this study, we identified a set of 73 miRNA markers differentially expressed between normal lung tissues and lung cancer tumors. This set of 73 miRNAs can accurately distinguish lung cancer tumors from normal lung cancer tissues with an overall accuracy greater than 92% in both supervised and unsupervised classification in separate patient cohorts (*n* = 484). Furthermore, seven miRNAs showed significant differential expression in lung cancer patient blood samples versus normal samples (Figure 5C). These seven miRNAs can accurately separate blood samples from lung cancer patients (*n* = 4) and normal individuals (*n* = 6) with 100% of accuracy in unsupervised clustering, indicating its potential utility as a minimally invasive diagnostic test. Among the seven miRNAs, hsa-miR-543 had concordant over-expression in both lung cancer tumor tissues and blood samples. Numerous miRNAs showed consistent expression patterns in NSCLC, SCLC, and carcinoid compared with normal lung tissues (Figure 4A,B). The majority of miRNAs showed consistent expression patterns between NSCLC and SCLC, but not with carcinoid when compared with normal lung tissues (Figure 4C). MiRNA-205, miRNA-363, and miRNA-99a had opposite expression patterns between NSCLC and SCLC in comparison with normal lung tissues. Among the identified 73 miRNAs, numerous miRNAs were consistently dysregulated in lung cancer tumors vs. normal lung tissues across different patient cohorts (Figure 5), 14 of which also had a consistent association with TCGA NSCLC patient survival outcomes (Table 2). Given the heterogeneity of lung cancer, we acknowledge that the identified 73 diagnostic miRNAs need to be further validated in separate patient cohorts to substantiate their clinical utility. Our study showed the feasibility to extract miRNAs from blood samples and use the expression of our identified miRNAs to distinguish four lung cancer patients from six normal individuals. We plan to validate the results using blood samples collected from more patients including the NLST in our future research.

Ten miRNAs were identified as potential tumor suppressors, including hsa-miR-144, hsa-miR-195, hsa-miR-223, hsa-miR-30a, hsa-miR-30b, hsa-miR-30d, hsa-miR-335, hsa-miR-363, hsa-miR-451, and hsa-miR-99a. Four miRNAs were identified as potential oncogenic, including hsa-miR-21, hsa-miR-31, hsa-miR-411, and hsa-miR-494 (Table 2). Hsa-miR-144 functions as a tumor suppressor by inhibiting epithelial-to-mesenchymal transition (EMT), proliferation, migration, invasion, metastasis, and angiogenesis in multiple human cancers [56,57,58,59,60,61,62,63]. Synergistically, the miR-144/451a cluster inhibits metastasis [64] and is a tumor suppressor in oral squamous cell carcinoma through the inhibition of cancer cell invasion, migration, and clonogenic potential [65], with potential therapeutic applications using biomimetic nanosystems [66]. Downregulation of miR-195 was reported in multiple human cancer types and linked to its tumor-suppressive or oncogenic functional roles [67]. MiR-195 was panresistant to 21 NCCN-recommended NSCLC drugs (Table 3). MiR-233 controls innate immune responses to maintain the homeostasis of myeloid cells [68] and blocks cell-cyle progression in myeloid cells [69] functioning through negative feedback loops. MiR-233 inhibits the proliferation and metastasis of oral squamous cell carcinoma [70] and NSCLC [71]. In other contexts, miR-233 was upregulated in pancreatic cancer, gastric cancer, and ovarian cancer and promoted cancer cell proliferation, migration, and invasion [72,73,74]. MiR-30a functions as a tumor suppressor in colon cancer [75], hepatocellular carcinoma [76], and breast cancer [77]. MiR-30b suppresses cancer cell growth, migration, and invasion in esophageal cancer [78] and papillary thyroid cancer [79]. MiR-30d suppresses NSCLC tumor invasion and migration by targeting Nuclear factor I B (*NFIB*) [80]. Although the tumor-suppressive roles of the miR-30 family have been reported in many human cancers, including lung cancer [81], miR-30 can also disrupt senescence and promote cancer through the inhibition of *p16^INK4A^* and *p53* [82]. Our analysis showed that the miR-30 family is panresistant to 21 NCCN-recommended drugs for treating NSCLC (Table 3). MiR-335 suppresses the proliferation of NSCLC cells by targeting *Tra2β* [83] and *CCNB2* [84]. MiR-363-3p inhibits proliferation and colony formation by targeting *PCNA* in lung adenocarcinoma A549 and H441 cells [85]. MiR-363-3p suppresses migration, invasion, and EMT by targeting *NEDD9* and *SOX4* in NSCLC [86]. MiR-451 inhibits NSCLC tumor cell growth and migration by targeting *LKB1/AMPK* [87] and *ATF2* [88] and sensitizes NSCLC cells to cisplatin by regulating *Mcl-1* [89]. MiR-99a suppresses EMT and stemness through the inhibition of two oncogenic proteins, E2F2 and EMR2 [90], and enhances radiation sensitivity by targeting mTOR in NSCLC [91]. All the evidence in the literature confirmed the 10 putative tumor suppressive miRNAs identified in our study. MiR-21 is a well-recognized oncogene in many human cancers, including NSCLC [92]. MiR-31 is also a prominent oncogene in lung cancer. Over-expression of miR-31 was found in major histological subtypes of NSCLC but not in SCLC or carcinoid in a published study [92]. In our patient cohort, significant over-expression of miR-31 and miR-31* was found in NSCLC and SCLC, but the results were not statistically significant in carcinoid when compared with normal lung tissues (Figure 4A). MiR-411 is oncogenic in lung cancer through the miR-411-SPRY4-AKT axis [93]. Tumor-derived miR-494 promotes angiogenesis in NSCLC under hypoxic conditions [94]. CircVAPA accelerates SCLC progression through the miR-377-3p and miR-494-3p/IGF1R/AKT axis [95]. All the published results confirmed the four oncogenic miRNAs identified in this study.

To decipher miRNA-mediated molecular machinery and determine the clinical relevance in NSCLC, we identified experimentally validated target genes of 73 miRNAs using TarBase. From these target genes, we identified pansensitive or panresistant genes to 21 NCCN-recommended drugs for treating NSCLC concordant at both mRNA and protein expression levels (Table 3). Genes associated with radiotherapy response in TCGA NSCLC patients were also identified (Appendix A). Among pansensitive genes, DGKE mRNA and protein expression was associated with sensitivity to erlotinib and was not associated with resistance to any of the 21 NCCN-recommended drugs (Figure 6A). Among panresistant genes, WDR47 mRNA and protein expression was associated with resistance to afatinib, brigatinib, and osimertinib (Figure 6B–D). *DGKE* and *WDR47* were both associated with radiotherapy sensitivity (Figure 7). Non-silent mutations of *DGKE* and *WDR47* were found in NSCLC cell lines. Recessive mutations in *DGKE* cause small-vessel thrombosis and kidney failure [96]. The silenced status of *DGKE* was found in nonadherent human colon cancer HT-29 cells [97]. DGKE was targeted by hsa-miR-3065-5p, which was over-expressed in colon cancer [98]. In this study, hsa-miR-3065-5p was under-expressed in lung cancer tumors. *WDR47* plays an important role in microtubule formation [99]. *WDR47* is targeted by hsa-miR-139-5p, which was found to be under-expressed in lung cancer tumors in this study. Our results provided new insight into hsa-miR-3065-5p/*DGKE* and hsa-miR-139-5p/*WDR47* in NSCLC. Further experimental investigation of these two genes will be conducted in our future research.

Based on the identified miRNA-mediated transcriptional networks in NSCLC, we sought to discover new drugs or new indications of existing drugs for treating NSCLC that were not known before. Specifically, among the target genes of the 73 diagnostic miRNAs, protective genes were selected as associated with favorable patient outcomes and under-expressed in lung cancer tumors; hazard genes were selected as associated with poor patient outcomes and over-expressed in lung cancer tumors. Proliferation genes were also selected from CRISPR-Cas9/RNAi screening data in NSCLC cell lines. Candidate compounds that can upregulate protective genes and downregulate hazard genes and proliferation genes were identified using CMap. Next, therapeutic compounds that are effective in inhibiting NSCLC cell growth are pinpointed from the candidate compounds. New indications for treating NSCLC were discovered for three drugs in this study, including BX-912, daunorubicin, and midostaurin. These findings provided evidence to design future clinical trials to test the efficacy of these three compounds as repositioning drugs in treating NSCLC. Most clinical trials failed because drug responders were not well-characterized. To characterize responders of these three potential repositioning drugs, genes associated with the response to these compounds were selected on a genome-wide scale in CCLE NSCLC cell lines (Table 4). The selected genes had a concordant significantly differential mRNA and protein expression in sensitive vs. resistant NSCLC cell lines to the specific compound. The majority of these genes had non-silent mutations and/or gene fusions in the examined NSCLC cell lines. The results presented in this study will accelerate the design of future clinical trials and expedite the R&D of repositioning drugs to improve lung cancer patient survival outcomes.

## 5. Conclusions

This study identified a set of 73 miRNAs for the accurate detection of lung cancer tumors from normal lung tissues. Seven miRNAs were also identified to classify blood samples from lung cancer patients from healthy individuals. Combined with survival analysis of TCGA NSCLC patients, 10 miRNAs were identified as potential tumor suppressors, and 4 as potential oncogenes, supported by their reported roles in published literature. From experimentally validated target genes of these 73 miRNAs, DGKE and WDR47 were found with significant associations with responses to both systemic therapies and radiotherapy in NSCLC. Based on our identified miRNA-mediated molecular network, BX-912, daunorubicin, and midostaurin were discovered as potential repositioning drugs for treating NSCLC. The findings from this study have implications for improving lung cancer diagnosis, optimizing treatment selection, and discovering new drug options for better patient outcomes.

## 6. Patents

The results in this paper are included in a US provisional patent application with Serial No. 63/494,623.

## Figures and Tables

**Figure 1 cancers-15-02294-f001:**
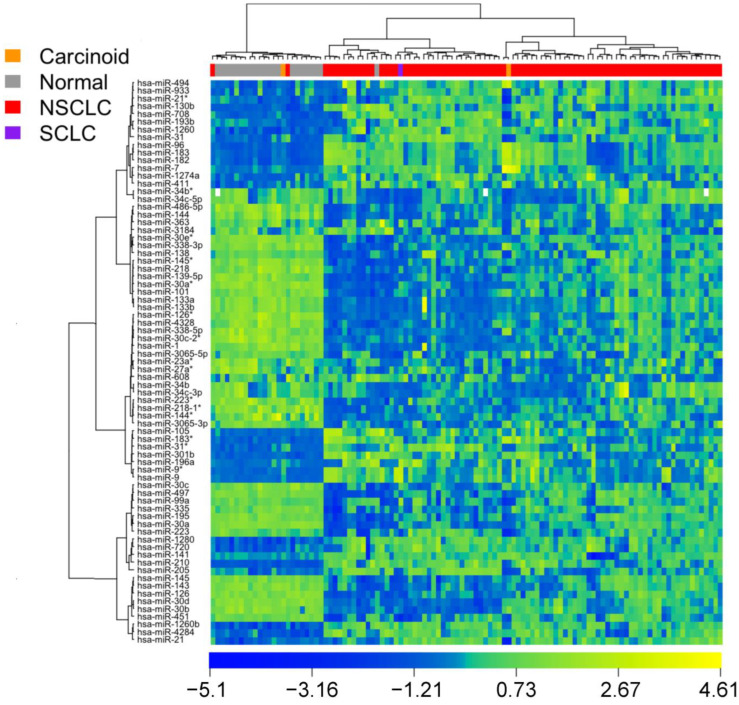
Heat map of miRNA-based clustering of patient tissue samples in the training set (MBRCC/CHTN; FDR < 0.05 in SAM, fold change > 2 or < 0.5; unpaired or paired *t*-tests). The asterisks in the miRNA annotations were provided by the manufacturer.

**Figure 2 cancers-15-02294-f002:**
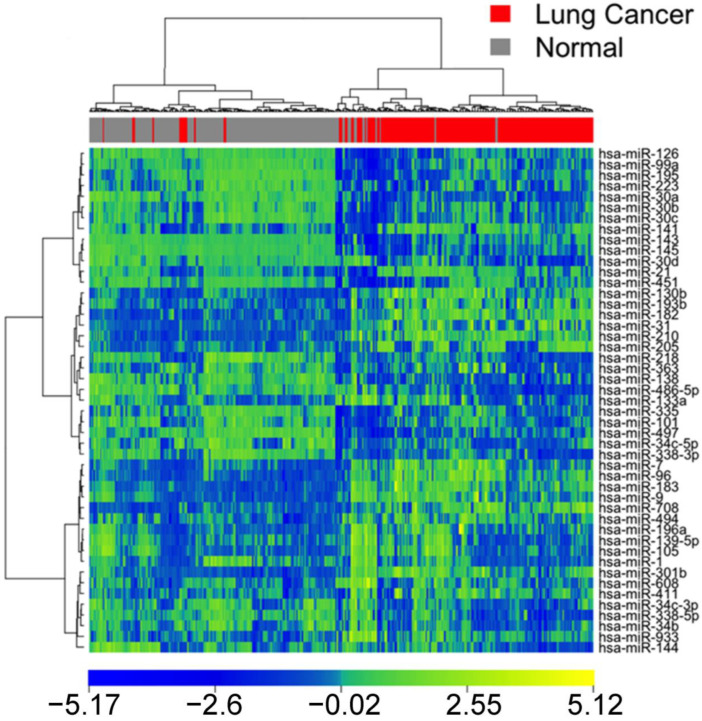
Heat map of miRNA-based clustering of patient samples in GSE15008.

**Figure 3 cancers-15-02294-f003:**
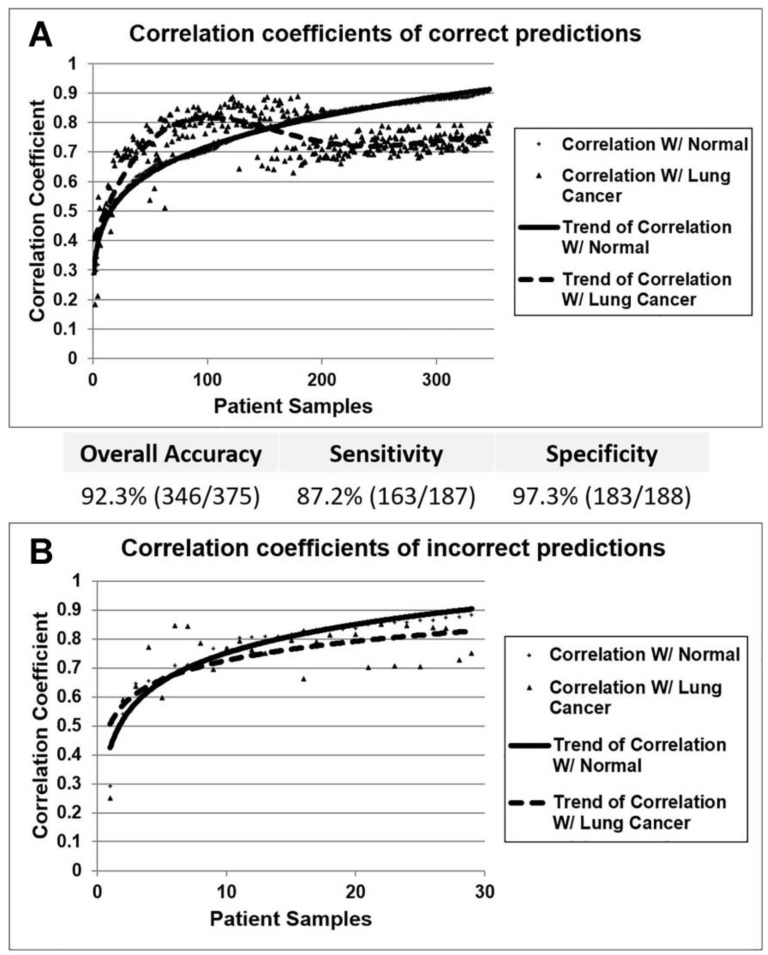
Supervised prediction of lung cancer based on expression profiles of the identified miRNA markers using nearest centroid classification. (**A**) Person’s correlation coefficients of correct predictions using nearest centroid algorithm. The prediction performance is shown in the table. The trend of correlation with the normal centroid was fitted with an exponentiation function, whereas the trend of correlation with the lung cancer centroid was fitted with a polynomial function. (**B**) Person’s correlation coefficients of incorrect predictions in the nearest centroid classification. The trend of the correlations was fitted with logarithmic functions.

**Figure 4 cancers-15-02294-f004:**
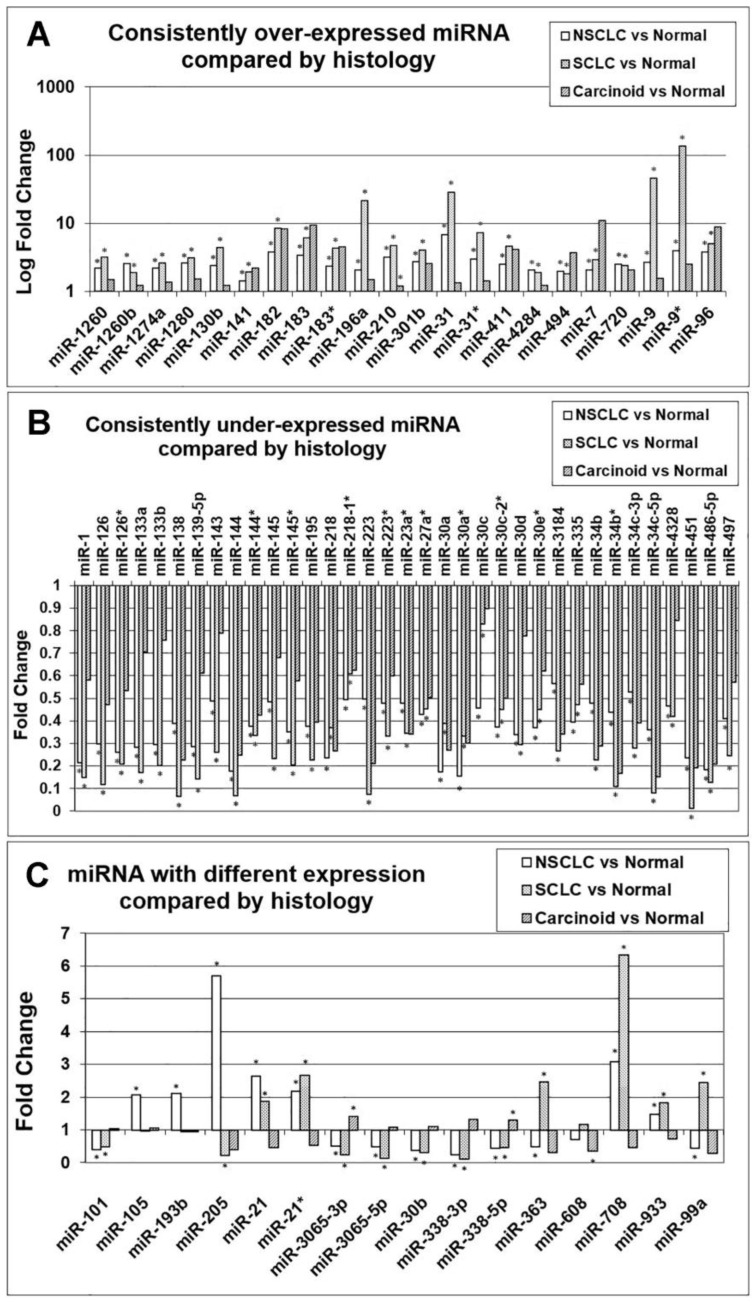
Expression patterns of the identified miRNA in different histology of lung cancer. (**A**) Consistently over-expressed miRNA in different histology. (**B**) Consistently under-expressed miRNA in different histology. (**C**) miRNA showing different expression patterns compared to histology. * Statistically significant at *p <* 0.05 in unpaired *t*-tests.

**Figure 5 cancers-15-02294-f005:**
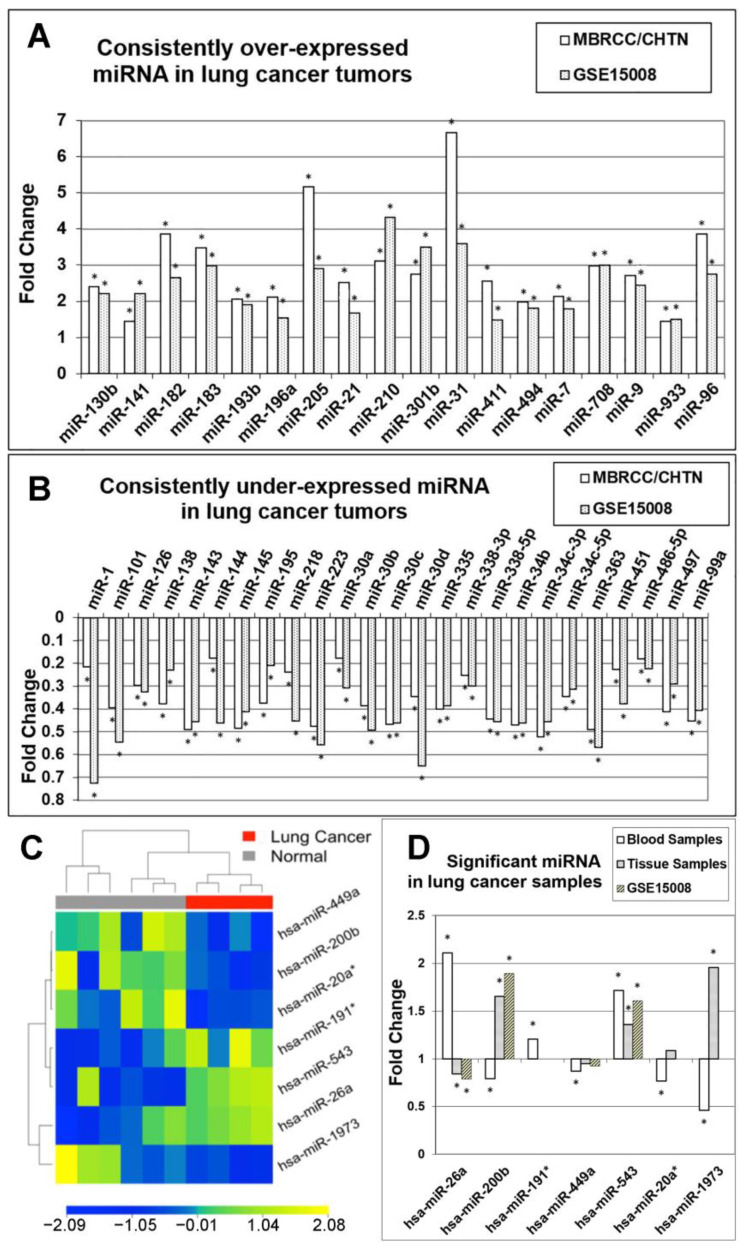
miRNA markers with a consistent expression of tumor versus normal samples in training and validation cohorts. (**A**) Consistently over-expressed miRNA in lung cancer tumors in MBRCC/CHTN and GSE15008. (**B**) Consistently under-expressed miRNA in lung cancer tumors compared with GSE15008. (**C**) Heatmap of significant genes found by regular *t*-tests on blood samples. (**D**) Expression patterns of 7 significant miRNA markers in lung cancer blood samples, tumor tissue samples in MBRCC/CHTN, and GSE15008 compared with normal samples in each cohort. “Tissue samples” refer to the samples in the MBRCC/CHTN cohort. * Statistically significant at *p <* 0.05 in unpaired *t*-tests.

**Figure 6 cancers-15-02294-f006:**
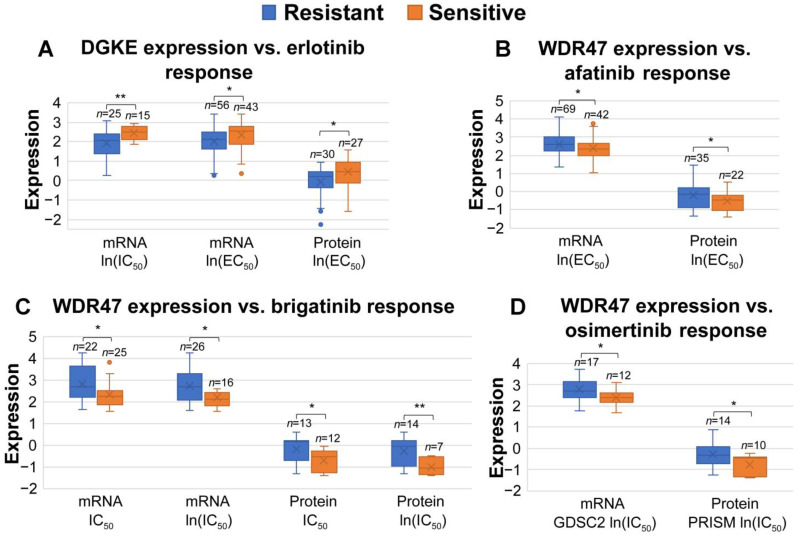
The association of mRNA/protein expression of DGKE and WDR47 with the responses to selected drugs in NSCLC cell lines. (**A**) DGKE was expressed significantly higher in erlotinib-sensitive NSCLC cell lines when categorized by ln(IC_50_) and ln(EC_50_) measurements in the PRISM data. (**B**) WDR47 was expressed significantly higher in afatinib-resistant NSCLC cell lines when categorized by ln(EC_50_) in the PRISM data. (**C**) WDR47 was expressed significantly higher in brigatinib-resistant NSCLC cell lines when categorized by IC_50_ and ln(IC_50_) measurements in the PRISM data. (**D**) WDR47 was expressed significantly higher in osimertinib-resistant NSCLC cell lines when categorized by ln(IC_50_) in the GDSC2 and PRISM data. * *p* < 0.05; ** *p* < 0.01.

**Figure 7 cancers-15-02294-f007:**
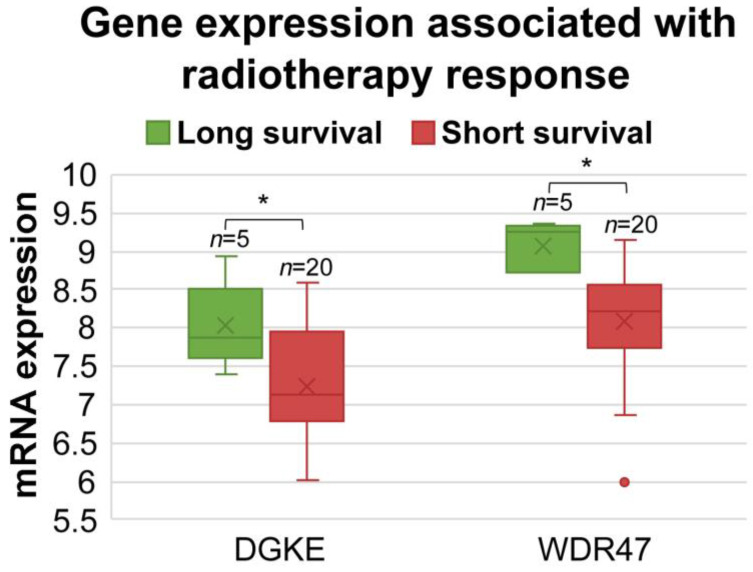
The mRNA expression of *DGKE* and *WDR47* was associated with radiotherapy sensitivity in TCGA-LUAD and TCGA-LUSC stage III and IV patients who had received radiotherapy. Long survival was defined as patients surviving longer than 58 months; short survival was defined as patients surviving shorter than 20 months. *DGKE* and *WDR47* were both expressed significantly higher in the long-survival patient group. * *p* < 0.05.

**Figure 8 cancers-15-02294-f008:**
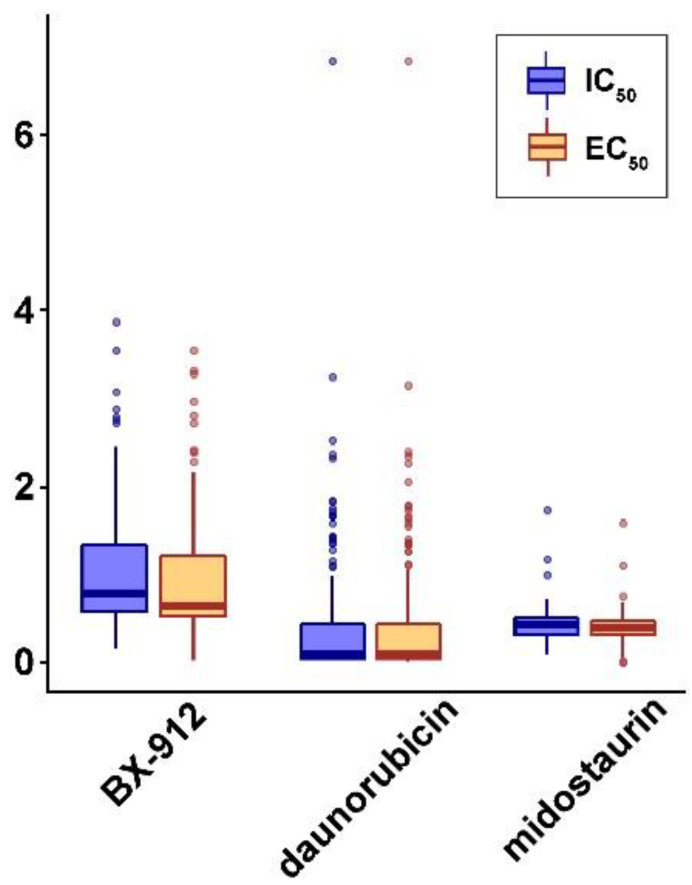
CMap selected compounds: BX-912, daunorubicin, and midostaurin, with a low average drug concentration (IC_50_ and EC_50_) in the CCLE NSCLC cell lines.

**Table 1 cancers-15-02294-t001:** Patient clinical information, including AJCC cancer stage, age group, gender, MBR tumor grade, and tissue histology.

Clinical Information	*N*
Cancer Stage	109
1	40
2	34
3	13
Normal	22
Age	
<60	23
60–80	68
>80	2
Missing	16
Gender	
Male	57
Female	51
Missing	1
Tumor Grade	
1	4
2	29
3	42
Normal	22
Missing	12
Histology	
NSCLC	
Adenocarcinoma	41
Adenosquamous carcinoma	1
Large cell carcinoma	2
Non-small cell carcinoma	13
Sarcomatoid carcinoma	1
Squamous cell carcinoma	1
Small Cell Carcinoma	1
Carcinoid tumor	2
Normal Tissue	22

**Table 2 cancers-15-02294-t002:** Potential tumor suppressive and potential oncogenic miRNAs.

Potential Tumor Suppressive miRNAs	Potential Oncogenic miRNAs
hsa-miR-144, hsa-miR-195, hsa-miR-223, hsa-miR-30a, hsa-miR-30b, hsa-miR-30d, hsa-miR-335, hsa-miR-363, hsa-miR-451, hsa-miR-99a	hsa-miR-21, hsa-miR-31, hsa-miR-411, hsa-miR-494

**Table 3 cancers-15-02294-t003:** Pansensitive and panresistant genes concordant in both mRNA and proteomics data, and pansensitive and panresistant miRNAs to 21 NCCN-recommended drugs for treating NSCLC. Blue font indicates the gene having at least one non-silent mutation in at least one NSCLC cell line. Red font indicates the gene having both fusion and non-silent mutation in NSCLC cell lines.

Drug	Pansensitive Genes	Panresistant Genes
afatinib	ADH7	CCNT1, WDR47, TRIB1, hsa-miR-133a, hsa-miR-1280
alectinib		CCNT1, hsa-miR-30b, hsa-miR-30d
brigatinib		WDR47, hsa-miR-133a, hsa-miR-30a
cabozantinib		hsa-miR-133a, hsa-miR-210
carboplatin		hsa-miR-210
cisplatin		CDK1, hsa-miR-133a, hsa-miR-30a, hsa-miR-30b
crizotinib	hsa-miR-223, hsa-miR-218	hsa-miR-30b, hsa-miR-30d
dabrafenib	hsa-miR-218	GLI2, CLCN5, CDKN3
dacomitinib		GLTP, FBXL4, hsa-miR-195, hsa-miR-30a
docetaxel		RAB30, NCEH1, GLTP, COPG1, hsa-miR-133a, hsa-miR-210, hsa-miR-30a
erlotinib	DGKE	GLTP, hsa-miR-1280
gefitinib		hsa-miR-195
gemcitabine		CDK1, CDK16
osimertinib		WDR47, hsa-miR-30a
paclitaxel		RAB30, hsa-miR-30a
pemetrexed		CDK16, hsa-miR-34b
trametinib		FOS, CDC42, PCBP1, TAOK2, NISCH, hsa-miR-195
vemurafenib		hsa-miR-195
vinorelbine		FOS, CDC42, RAB30, NCEH1, GLTP, hsa-miR-133a, hsa-miR-30a

**Table 4 cancers-15-02294-t004:** Genes associated with sensitivity/resistance to the selected compounds with concordant mRNA and protein expression. Blue font indicates the gene having at least one non-silent mutation in NSCLC cell lines. Red font indicates the gene having both fusions and non-silent mutations in NSCLC cell lines.

Drug	Sensitive Genes	Resistant Genes
BX-912	ARSD, ATG4C, CD44, CDK6, CNST, G3BP2, GSTA5, KDM6B, LGMN, MCM9, NCOR2, NEGR1, NPC2, NT5C, PTPRG, RAB32, RPTOR, RRBP1, SF3B5, SOGA3, STAU1, TGFB2, TMCC2	BCAS1, CD14, CGNL1, PALM3, VTCN1
daunorubicin	BCOR, KMT2B, NUDT8, OTUD4, SMS, SYT7	
midostaurin	ACBD5, APLF, ASH1L, BSCL2, CENPB, FMO5, JAK1, NRCAM, OPA3, PCGF1, SHC1, WDR53, ZHX3	

## Data Availability

All data are publicly available with access information provided in the manuscript.

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
