# Peer review of "MicroRNA, mRNA, and Proteomics Biomarkers and Therapeutic Targets for Improving Lung Cancer Treatment Outcomes"

_cancers, 2023, doi:10.3390/cancers15082294_

Round 1
Reviewer 1 Report
The article touches upon very important and topical issues in the diagnosis and sensitivity to chemotherapy of non-small cell lung cancer. The authors analyzed and presented in the article a very large amount of data, which somewhat complicates the perception of the material. It is difficult for the reader to keep track of what part of the work was done on real lung cancer samples, and what part was done on datasets obtained from open databases.
The Introduction is written too short, and the Methods, on the contrary, are too detailed in places. If the authors consider such a detailed description of the theoretical foundations of the methods to be important, then perhaps it should be included in the Introduction.
Minor comments.
Given the heterogeneity of the non-small cell lung cancer samples, we would like to see more samples analyzed.
Specify the criteria for the quality of the isolated RNA.
Line 145. Check for typos. "and spiking" is repeated twice.
The norms were taken from patients with lung cancer. Whether any control was carried out to confirm that there are no tumor cells in these samples?
Table 1 might be better moved to point 2.1.
Mayor comments.
Very few blood samples have been collected. If the authors consider this part of the study important, then the collection of blood samples should be increased at least 10 times. If it is impossible to increase the collection (or it is technically difficult to analyze miRNAs in it), then the part concerning the analysis of microRNAs in blood should be deleted from the article.
Lines 371, 373. As far as I understand, we are talking about the logarithm of the ratio of expression values. This must be specified. Also indicate which logarithm was used (decimal, normal, etc.).
Lines 443, 445. Same remark.
Lines 508-509. Explain what "The false positive rate of CT screening 508 was about 96.4%" means. Too many false positives.
The article has a lot of discussion about the DGKE and WDR47 genes. Why was their expression not determined in experimental samples? It should be.
Author Response
Reviewer 1
Comments and Suggestions for Authors
Reviewer: The article touches upon very important and topical issues in the diagnosis and sensitivity to chemotherapy of non-small cell lung cancer. The authors analyzed and presented in the article a very large amount of data, which somewhat complicates the perception of the material. It is difficult for the reader to keep track of what part of the work was done on real lung cancer samples, and what part was done on datasets obtained from open databases.
The Introduction is written too short, and the Methods, on the contrary, are too detailed in places. If the authors consider such a detailed description of the theoretical foundations of the methods to be important, then perhaps it should be included in the Introduction.
Authors: We thank the reviewer for the constructive comments. We have revised the Introduction and Methods accordingly. The data sources are also specified in the Introduction and the manuscript.
Minor comments.
Reviewer: Given the heterogeneity of the non-small cell lung cancer samples, we would like to see more samples analyzed.
Authors: In this study, the diagnostic capacity of the identified miRNAs was validated in 462 patient samples. The prognostic capacity of the miRNAs was evaluated in 1,016 non-small cell lung cancer (NSCLC) samples. The drug responses of miRNA, mRNA, and protein markers were investigated in 135 human NSCLC cell lines. Given the heterogeneity of lung cancer, we acknowledge that the identified 73 diagnostic miRNAs need to be further validated in separate patient cohorts to substantiate their clinical utility. This is now added to the Discussion.
Reviewer: Specify the criteria for the quality of the isolated RNA.
Authors: RNA quality was checked with UV Spectrometry (260/280 > 1.8) and RNA quality determination gel (1% agarose – 2% formaldehyde QC gel). It is now added to Section 2.2.
Reviewer: Line 145. Check for typos. "and spiking" is repeated twice.
Authors: Corrected
Reviewer: The norms were taken from patients with lung cancer. Whether any control was carried out to confirm that there are no tumor cells in these samples?
Authors: The samples were examined by pathologists before being cataloged in the biorepositories at West Virginia University and Cooperative Human Tissue Network (CHTN) operated by the National Cancer Institute. For tumor tissue samples, it was certified that the tumor content is at least 50% in each sample. For non-cancerous normal lung tissues, it was confirmed that no tumor was present in the sample. This information was provided in the pathology reports of the tissue samples that we received. Before our experiments, a pathologist’s assistant further examined the tissue samples. No experimental control was carried out to confirm that there are no tumor cells in the normal tissues. It is now added to Section 2.1.
Reviewer: Table 1 might be better moved to point 2.1.
Authors: We now move Table 1 to Section 2.1.
Mayor comments.
Reviewer: Very few blood samples have been collected. If the authors consider this part of the study important, then the collection of blood samples should be increased at least 10 times. If it is impossible to increase the collection (or it is technically difficult to analyze miRNAs in it), then the part concerning the analysis of microRNAs in blood should be deleted from the article.
Authors: Due to time and budget constraints, we only analyzed 10 blood samples in this study. Our study showed the feasibility to extract miRNAs from blood samples and use the expression of our identified miRNAs to distinguish 4 lung cancer patients from 6 normal individuals. We plan to validate the results using blood samples collected from more patients including the National Lung Screening Trial (NLST) in our future research. This is now added to the Discussion.
Reviewer: Lines 371, 373. As far as I understand, we are talking about the logarithm of the ratio of expression values. This must be specified. Also indicate which logarithm was used (decimal, normal, etc.).
Authors: In this study, the miRNA, mRNA, and protein expression values are log2-transformed for further analysis. The hazard ratios are obtained from univariate Cox modeling. It is now specified in Materials and Methods and Results.
Reviewer: Lines 443, 445. Same remark.
Authors: In this study, the miRNA, mRNA, and protein expression values are log2-transformed for further analysis. The hazard ratios are obtained from univariate Cox modeling. It is now specified in Materials and Methods and Results.
Reviewer: Lines 508-509. Explain what "The false positive rate of CT screening 508 was about 96.4%" means. Too many false positives.
Authors: According to NLST, The false positive rate of CT screening was about 96.4% [1]. Thus, 96 out of 100 individuals who undergo CT screenings will have a false-positive result showing the presence of a lung nodule or another abnormality that is not cancer. This is now added to the Discussion.
Reviewer: The article has a lot of discussion about the DGKE and WDR47 genes. Why was their expression not determined in experimental samples? It should be.
Authors: DGKE and WDR47 genes were found as targeted genes of our identified diagnostic miRNAs. Their clinical implications were revealed from public data TCGA and CCLE. Further experiments on these two genes will be conducted in our future research. It is now added to the Discussion.
References
- Aberle, D.R., et al., Reduced lung-cancer mortality with low-dose computed tomographic screening. N Engl J Med, 2011. 365(5): p. 395-409.

Reviewer 2 Report
Overall, this paper is writing and presenting well, the data set are presented appropriately, and the discussion matches the aim of the study.
I have minor suggestion:
Line 30:
I assume pan sensitive should be written pan sensitive.
Line 145:
and spiking was written twice.
Line 355-357:
Some repetitions of ‘’in lung cancer blood samples’’, I think it could be better to rephrase the paragraph.
Line 419-421:
Here also, it would be better to rephrase the paragraph.
Author Response
Reviewer 2
Comments and Suggestions for Authors
Reviewer: Overall, this paper is writing and presenting well, the data set are presented appropriately, and the discussion matches the aim of the study.
Authors: We appreciate the very positive comments from the reviewer.
I have minor suggestion:
Line 30:
Reviewer: I assume pan sensitive should be written pan sensitive.
Authors: Corrected
Reviewer: Line 145:
and spiking was written twice.
Authors: Corrected
Reviewer: Line 355-357:
Some repetitions of ‘’in lung cancer blood samples’’, I think it could be better to rephrase the paragraph.
Authors: We revised this paragraph to reduce the repetitions.
Reviewer: Line 419-421:
Here also, it would be better to rephrase the paragraph.
Authors: We paraphrased this paragraph.

Reviewer 3 Report
Ye et al. show the relevance of set of miRNAs as biomarkers in drug responsiveness and lung cancer treatment outcomes.
This paper is well designed to substantiate the findings.
However, these comments will help in the better impact of this paper.
1. The authors should emphasize the purpose of study to explore (hsa-miR-144 in lung cancer prognosis. Since, these miRNAs such as hsa-miR-144 is also reported to implicated with other cancer types such as oral cancer.
2. The authors defend the relevance of intracellular miRNAs and exosomal miRNAs separately.
3. The authors may discuss the relevance of lung cancer heterogeneity and probability of miRNAs as biomarkers.
4. The authors explore the uses of AI and machine learning in the proposed miRNAs as biomarkers.
5. The authors may discuss the suitability of miRNAs as biomarkers and various biological fluids and materials.
Author Response
Reviewer 3
Comments and Suggestions for Authors
Reviewer: Ye et al. show the relevance of set of miRNAs as biomarkers in drug responsiveness and lung cancer treatment outcomes.
This paper is well designed to substantiate the findings.
Authors: We thank the reviewer for the positive comments.
However, these comments will help in the better impact of this paper.
Reviewer: 1. The authors should emphasize the purpose of study to explore (hsa-miR-144 in lung cancer prognosis. Since, these miRNAs such as hsa-miR-144 is also reported to implicated with other cancer types such as oral cancer.
Authors: Hsa-miR-144 was found as a potential tumor suppressor in lung cancer in this study. Hsa-miR-144 functions as a tumor suppressor by inhibiting epithelial-to-mesenchymal transition (EMT), proliferation, migration, invasion, metastasis, and angiogenesis in multiple human cancers [1-8]. Synergistically, the miR-144/451a cluster inhibits metastasis [9] and is a tumor suppressor in oral squamous cell carcinoma through the inhibition of cancer cell invasion, migration, and clonogenic potential [10], with potential therapeutic applications using biomimetic nanosystems [11]. This is now added to the Discussion.
Reviewer: 2. The authors defend the relevance of intracellular miRNAs and exosomal miRNAs separately.
Authors: Intracellular and exosomal miRNAs play different roles in maintaining cellular homeostasis and intercellular communication. Intracellular miRNAs attach to particular mRNA molecules and prevent their translation within cells [12]. Intracellular miRNAs play a role in a variety of biological functions, including development, differentiation, metabolism, and diseases including cancer, cardiovascular and neurodegenerative disorders [13]. In contrast, exosomal miRNAs are enclosed within exosomes, which are tiny vesicles that are released by cells. MiRNAs and other proteins can be transported between cells by exosomes, which are essential for intercellular communication [14]. Exosomal miRNAs control gene expression in target cells and participate in many biological processes, including immune control, angiogenesis, and cancer metastasis [15], and can also serve as diagnostic and prognostic biomarkers for various diseases, including cancer [16], cardiovascular disease, and infectious diseases [17]. This is now added to the Introduction.
Reviewer: 3. The authors may discuss the relevance of lung cancer heterogeneity and probability of miRNAs as biomarkers.
Authors: Lung cancer is a heterogeneous disease with distinct histological subtypes and complex somatic mutations. Tumor heterogeneity within disease sites contributes to resistance to cancer therapies and poses a challenge to biomarker discovery [50, 51]. MiRNAs have emerged as important diagnostic and prognostic biomarkers and therapeutic targets in cancer treatment due to their functional involvement in gene and protein regulation and intracellular signaling [52, 53]. MiRNA biomarkers are promising for the early detection of NSCLC supplementing low-dose CT screening [54] and the prognosis of advanced NSCLC patients treated with immunotherapy [55]. This is now added to the Discussion.
Reviewer: 4. The authors explore the uses of AI and machine learning in the proposed miRNAs as biomarkers.
Authors: The highly conserved family of tissue-specific miRNAs keeps cells in a stable state by negatively regulating gene expression in general. Since intracellular and extracellular miRNAs have a broad range of target genes and affect almost every signaling pathway, from cell cycle checkpoints to cell proliferation to apoptosis, proper regulation of miRNA expression is necessary to maintain normal physiology. Some miRNAs function as tumor suppressors and oncogenes, and their expression is dysregulated in different cancers. Although cancer treatments are currently available to slow the growth and spread of tumors, there aren't many effective diagnostic and treatment methods for different cancers. Specific miRNA profiling can distinguish molecularly diverse tumors based on their phenotypic characteristics, which can then be used to overcome diagnostic and therapeutic obstacles [18]. The available artificial intelligence/machine learning (AI/ML) tools, bioinformatics resources, and data consortia accelerate the discoveries of miRNA-based theranostics. This is now added to the Introduction.
Reviewer: 5. The authors may discuss the suitability of miRNAs as biomarkers and various biological fluids and materials.
Authors: The presence of miRNAs in a variety of bodily fluids, such as serum, plasma, saliva, and amniotic fluid [18], is a particularly fascinating but little-understood aspect of miRNA biology. Diagnostic miRNA markers were identified in tissues [19, 20], serum [21], blood [22], and sputum samples [23] from NSCLC patients. It is now added to Introduction.
References
- Pan, Y., et al., miR-144 functions as a tumor suppressor in breast cancer through inhibiting ZEB1/2-mediated epithelial mesenchymal transition process. Onco Targets Ther, 2016. 9: p. 6247-6255.
- Yin, Y., et al., MiR-144 suppresses proliferation, invasion, and migration of breast cancer cells through inhibiting CEP55. Cancer Biology & Therapy, 2018. 19(4): p. 306-315.
- Kooshkaki, O., et al., MiR-144: A New Possible Therapeutic Target and Diagnostic/Prognostic Tool in Cancers. Int J Mol Sci, 2020. 21(7).
- Sheng, S., et al., MiR-144 inhibits growth and metastasis in colon cancer by down-regulating SMAD4. Biosci Rep, 2019. 39(3).
- Sun, X.B., et al., MicroRNA-144 Suppresses Prostate Cancer Growth and Metastasis by Targeting EZH2. Technol Cancer Res Treat, 2021. 20: p. 1533033821989817.
- Wu, M., et al., MicroRNA-144-3p suppresses tumor growth and angiogenesis by targeting SGK3 in hepatocellular carcinoma. Oncol Rep, 2017. 38(4): p. 2173-2181.
- Gu, J., et al., MicroRNA-144 inhibits cell proliferation, migration and invasion in human hepatocellular carcinoma by targeting CCNB1. Cancer Cell Int, 2019. 19: p. 15.
- Chen, S., et al., MiR-144 inhibits proliferation and induces apoptosis and autophagy in lung cancer cells by targeting TIGAR. Cell Physiol Biochem, 2015. 35(3): p. 997-1007.
- Zhang, J., et al., Transcriptional control of PAX4-regulated miR-144/451 modulates metastasis by suppressing ADAMs expression. Oncogene, 2015. 34(25): p. 3283-95.
- Manasa, V.G., S. Thomas, and S. Kannan, MiR-144/451a cluster synergistically modulates growth and metastasis of Oral Carcinoma. Oral Dis, 2023. 29(2): p. 584-594.
- Li, K., et al., Biomimetic Nanosystems for the Synergistic Delivery of miR-144/451a for Oral Squamous Cell Carcinoma. Balkan Med J, 2022. 39(3): p. 178-186.
- Turchinovich, A. and B. Burwinkel, Distinct AGO1 and AGO2 associated miRNA profiles in human cells and blood plasma. RNA Biol, 2012. 9(8): p. 1066-75.
- Bartel, D.P., Metazoan MicroRNAs. Cell, 2018. 173(1): p. 20-51.
- Valadi, H., et al., Exosome-mediated transfer of mRNAs and microRNAs is a novel mechanism of genetic exchange between cells. Nat Cell Biol, 2007. 9(6): p. 654-9.
- Zhang, Y., et al., Exosomes: biogenesis, biologic function and clinical potential. Cell & Bioscience, 2019. 9(1): p. 19.
- Wang, M., et al., Emerging Function and Clinical Values of Exosomal MicroRNAs in Cancer. Mol Ther Nucleic Acids, 2019. 16: p. 791-804.
- Mendell, J.T. and E.N. Olson, MicroRNAs in stress signaling and human disease. Cell, 2012. 148(6): p. 1172-87.
- Cortez, M.A., et al., MicroRNAs in body fluids--the mix of hormones and biomarkers. Nat Rev Clin Oncol, 2011. 8(8): p. 467-77.
- Bishop, J.A., et al., Accurate Classification of Non–Small Cell Lung Carcinoma Using a Novel MicroRNA-Based ApproachMicroRNA-Based Approach for Lung Cancer Classification. Clinical Cancer Research, 2010. 16(2): p. 610-619.
- Boeri, M., et al., MicroRNA signatures in tissues and plasma predict development and prognosis of computed tomography detected lung cancer. Proceedings of the National Academy of Sciences, 2011. 108(9): p. 3713-3718.
- Chen, X., et al., Identification of ten serum microRNAs from a genome-wide serum microRNA expression profile as novel noninvasive biomarkers for nonsmall cell lung cancer diagnosis. Int J Cancer, 2012. 130(7): p. 1620-8.
- Keller, A., et al., miRNAs in lung cancer-studying complex fingerprints in patient's blood cells by microarray experiments. BMC cancer, 2009. 9(1): p. 1-10.
- Yu, L., et al., Early detection of lung adenocarcinoma in sputum by a panel of microRNA markers. International journal of cancer, 2010. 127(12): p. 2870-2878.

Round 2
Reviewer 1 Report
Thank you for answering the comments.
Reviewer 3 Report
The authors have incorporated majority of suggestions.